# Overlapping Spaces for Compact Graph Representations

**Kirill Shevkunov**
Yandex, MIPT
Moscow, Russia
shevkunov.ks@phystech.edu

**Liudmila Prokhorenkova**
Yandex Research, MIPT, HSE University
Moscow, Russia
ostroumova-la@yandex.ru

## Abstract

Various non-trivial spaces are becoming popular for embedding structured data such as graphs, texts, or images. Following spherical and hyperbolic spaces, more general *product spaces* have been proposed. However, searching for the best configuration of a product space is a resource-intensive procedure, which reduces the practical applicability of the idea. We generalize the concept of product space and introduce an *overlapping space* that does not have the configuration search problem. The main idea is to allow subsets of coordinates to be shared between spaces of different types (Euclidean, hyperbolic, spherical). As a result, we often need fewer coordinates to store the objects. Additionally, we propose an optimization algorithm that automatically learns the optimal configuration. Our experiments confirm that overlapping spaces outperform the competitors in graph embedding tasks with different evaluation metrics. We also perform an empirical analysis in a realistic information retrieval setup, where we compare all spaces by incorporating them into DSSM. In this case, the proposed overlapping space consistently achieves nearly optimal results without any configuration tuning. This allows for reducing training time, which can be essential in large-scale applications.

## 1 Introduction

Building vector representations of various objects is one of the central tasks of machine learning. Word embeddings such as Glove [22] and Word2Vec [19] are widely used in natural language processing; a similar Prod2Vec [7] approach is used in recommendation systems. There are many algorithms proposed for graph embeddings, e.g., Node2Vec [8] and DeepWalk [23]. Recommendation systems often construct embeddings of a bipartite graph that describes interactions between users and items [10].

For a long time, embeddings were considered exclusively in $\mathbb{R}^n$. However, the hyperbolic space was shown to be more suitable for graph, word, and image representations due to the underlying hierarchical structure [12, 20, 21, 26]. Going beyond spaces of constant curvature, a recent study [9] proposed *product spaces*, which combine several copies of Euclidean, spherical, and hyperbolic spaces. While these spaces demonstrate promising results, the optimal signature (types of combined spaces and their dimensions) has to be chosen via brute force, which may not be acceptable in large-scale applications.

In this paper, we propose a more general metric space called *overlapping space* (OS) together with an optimization algorithm that trains signature *simultaneously* with embedding allowing us to avoid brute-forcing. The main idea is to allow coordinates to be shared between different spaces, which significantly reduces the number of coordinates needed.

35th Conference on Neural Information Processing Systems (NeurIPS 2021).

Importantly, the proposed overlapping space can further be enhanced by adding non-metric approaches such as *weighted inner product* [13] as additional similarity measures complementing metric ones. Thus, we obtain a flexible hybrid measure *OS-Mixed* that is no longer a metric space. Our experiments show that in some cases, non-metric measures outperform metric ones. The proposed OS-Mixed has advantages of both worlds and thus achieves superior performance.

To validate the usefulness of the proposed overlapping space, we provide an extensive empirical evaluation for the task of graph embedding, where we consider both distortion-based (i.e., preserving distances) and ranking-based (i.e., preserving neighbors) objectives. In both cases, the proposed measure outperforms the competitors. We also compare the spaces in information retrieval and recommendation tasks, for which we apply them to train embeddings via DSSM [11]. Our method works comparable to the best product spaces tested in these cases, while it does not require brute-forcing for the best signature. Thus, using the overlapping space may significantly reduce the training time, which can be crucial in large-scale applications.

## 2 Background and related work

### 2.1 Embeddings and loss functions

For a graph $G = (V, E)$ an embedding is a mapping $f : V \to U$, where $U$ is a metric space equipped with a distance $d_U : U \times U \to \mathbb{R}_+$.[1] On the graph, one can consider a shortest path distance $d_G : V \times V \to \mathbb{R}_+$. In the graph reconstruction task, it is expected that a good embedding preserves the original graph distances: $d_G(v, u) \approx d_U(f(v), f(u))$. The most commonly used evaluation metric is *distortion*, which averages relative errors of distance reconstruction over all pairs of nodes:

$$D_{avg} = \frac{2}{|V|(|V| - 1)} \sum_{(v,u) \in V^2, v \neq u} \frac{|d_U(f(v), f(u)) - d_G(v, u)|}{d_G(v, u)} . \tag{1}$$

While commonly used in graph reconstruction, distortion is a natural choice for many practical applications. For example, in recommendation tasks, one usually deals with a partially observed graph (some positive and negative element pairs), so a huge graph distance between two nodes in the observed part does not necessarily mean that the nodes are not connected by a short path in the full graph. Also, often only the order of the nearest elements is essential while predicting distances to faraway objects is not critical. In such cases, it is more reasonable to consider a local ranking metric, e.g., the mean average precision (mAP) that measures the relative closeness of the relevant (adjacent) nodes compared to the others:[2]

$$\mathrm{mAP} = \frac{1}{|V|} \sum_{v \in V} \mathrm{AP}(v) = \frac{1}{V} \sum_{v \in V} \frac{1}{\deg(v)} \sum_{u \in N_v} \frac{|N_v \cap R_v(u)|}{|R_v(u)|},$$
$$R_v(u) = \{w \in V | d_U(f(v), f(w)) \leq d_U(f(v), f(u))\}, \ N_v = \{w \in V | (v, w) \in E\}. \tag{2}$$

Mean average precision cannot be directly optimized since it is not differentiable. In our experiments, we use the following probabilistic loss function as a proxy:[3]

$$L_{proxy} = - \sum_{(v,u) \in E} \log \mathrm{P}((v, u) \in E) = - \sum_{(v,u) \in E} \log \frac{\exp(-d_U(f(v), f(u)))}{\sum_{w \in V} \exp(-d_U(f(v), f(w)))} . \tag{3}$$

Note that when substituting $d_U(x, y) = c - f(x)^T f(y)$ (assuming that $f(x) \in \mathbb{R}^n$, so the dot product is defined), $L_{proxy}$ becomes the standard word2vec loss function.

### 2.2 Spaces, distances, and similarities

In the previous section, we assumed that $d_U : U \times U \to \mathbb{R}_+$ is an arbitrary distance. In this section, we discuss particular choices often considered in the literature.

---

[1]Note that any discrete metric space corresponds to a weighted graph, so graph terminology is not restrictive.

[2]For mAP, the relevance labels are assumed to be binary (unweighted graphs). If a graph is weighted, then we say that $N_v$ consists of the closest element to $v$ (or several closest elements if the distances to them are equal).

[3]See Table 10 in Appendix for the comparison of other ways of converting distance to probability.

For many years, Euclidean space was the primary choice for structured data embeddings [6]. For two points $x, y \in \mathbb{R}^d$, Euclidean distance is defined as $d_E(x, y) = \left( \sum_{i=1}^{d} (x_i - y_i)^2 \right)^{1/2}$.

Spherical spaces were also found to be suitable for some applications [18, 24, 30]. Indeed, in practice, vector representations are often normalized, so cosine similarity between vectors is a natural way to measure their similarity. This naturally corresponds to a spherical space $S_d = \{x \in \mathbb{R}^{d+1} : \|x\|_2^2 = 1\}$ equipped with a distance $d_S(x, y) = \arccos(x^T y)$.

In recent years, hyperbolic spaces also started to gain popularity. Hyperbolic embeddings have shown their superiority over Euclidean ones in a number of tasks, such as graph reconstruction and word embedding [20, 21, 25, 26]. To represent the points, early approaches used the Poincaré model of the hyperbolic space [20], but later it has been shown that the hyperboloid (Lorentz) model may lead to more stable results [21]. In this work, we also adopt the hyperboloid model $H_d = \{x \in \mathbb{R}^{d+1} | \langle x, x \rangle_h = 1, x_1 > 0\}$ equipped with a distance $d_H = \operatorname{arccosh}(\langle x, y \rangle_h)$, where $\langle x, y \rangle_h := x_1 y_1 - \sum_{i=2}^{d+1} x_i y_i$.

Going even further, a recent paper [9] proposed more complex *product spaces* that combine several copies of Euclidean, spherical, and hyperbolic spaces. Namely, the overall dimension $d$ is split into $k$ parts (smaller dimensions): $d = \sum_{i=1}^{k} d_i$, $d_i > 0$. Each part is associated with a space $D_i \in \{E_{d_i}, S_{d_i}, H_{d_i}\}$ and scale coefficient $w_i \in \mathbb{R}_+$. Varying scale coefficients corresponds to changing curvature of hyperbolic and spherical spaces, while in Euclidean space this coefficient is not used ($w_i = 1$). Then, the distance in the product space $D_1 \times \ldots \times D_k$ is defined as:

$$d_P(x, y) = \sqrt{\sum_{i=1}^{k} w_i \, d_{D_i}(x[t_{i-1} + 1 : t_i], y[t_{i-1} + 1 : t_i])^2},$$

where $t_0 = 0$, $t_i = t_{i-1} + d_i$, and $x[s : e]$ is a subvector $(x_s, \ldots, x_e) \in \mathbb{R}^{e-s+1}$. If $k = 1$, we get a standard Euclidean, spherical, or hyperbolic space.

In [9], it is proposed to learn an embedding and scale coefficients $w_i$ simultaneously. However, choosing the optimal signature (how to split $d$ into $d_i$ and which types of spaces to choose) is challenging. A heuristics proposed in [9] allows to guess types of spaces if $d_i$'s are given. If $d_1 = d_2 = 5$, this heuristics agrees well with the experiments on three considered datasets. The generalizability of this idea to other datasets and configurations is unclear. In addition, it cannot be applied if a dataset is partially observed and graph distances cannot be computed (e.g., when there are several known positive and negative pairs). Hence, in practice, it is more reliable to choose a signature via the brute-force, which can be infeasible on large datasets.

Another way to measure objects' similarity, which is rarely compared with metric methods but is frequently used in practical applications, is via the dot product of vectors $x^T y$ or its weighted version $x^T W y$ with a diagonal matrix $W$, which is also known as a weighted inner product [13]. Such measures cannot be converted to a metric distance via a monotone transformation. However, they can be used to predict similarity or dissimilarity between objects, which is often sufficient in practice, especially when ranking metrics are used.

In this paper, we stress that when comparing different methods, both metric and non-metric variants should be used because different methods are better for different tasks. In particular, the dot product similarity allows one to easily differentiate between more popular and less popular items (the vector norm can be considered a measure of popularity). This feature is also attributed to hyperbolic spaces, where more popular items are placed closer to the origin.

## 2.3 Optimization

Gradient optimization in Euclidean space is straightforward, while for spherical or hyperbolic embeddings, we also have to control that the points belong to a surface. In previous works, Riemann-SGD was used to solve this problem [2]. In short, it projects Euclidean gradients on the tangent space at a point and then uses a so-called exponential map to move the point along the surface according

to the gradient projection. For product spaces, a generalization of the exponential map has been proposed [5, 27].

In [29], the authors compare RSGD with the retraction technique, where points are moved along the gradients in the ambient space and are projected onto the surface after each update. From their experiment, the retraction technique requires from 2% to 46% more iterations, depending on the learning rate. However, the exponential update step takes longer. Hence, the advantage of RSGD in terms of computation time depends on the specific implementation.

## 3  Overlapping spaces

### 3.1  Overlapping metric spaces

In this section, we propose a new concept of *overlapping spaces*.[4] This approach generalizes product spaces and allows us to make the signature (types and dimensions of combined spaces) trainable. Our main idea is to divide the embedding vector into several *overlapping* (unlike product spaces) segments, each segment corresponding to its own space. Then, instead of discrete signature brute-forcing, we optimize the weights of the signature elements.

Importantly, we allow the same coordinates of an embedding vector to be included in distances computations for spaces of different geometry. For this purpose, we first need to map a vector $x \in \mathbb{R}^d$ (for any $d \geq 1$) to a point in Euclidean, hyperbolic, and spherical space. Let us denote this mapping by $M$. Obviously, for Euclidean space, we may take $M_E(x) = x$. We may use the vector normalization for the spherical spaces, and for $H_d$ we use a projection from a hyperplane to a hyperboloid:

$$M_S(x) = \frac{x}{|x|} \in S_{d-1}, \ M_H(x) = \left( \sqrt{1 + \sum_{i=2}^{d} x_i^2}, x_1, \ldots, x_d \right) \in H_d. \tag{4}$$

Note that for such parametrization a $d$-dimensional vector $x$ is mapped into Euclidean and hyperbolic spaces of dimension $d$ and into a spherical space of dimension $d - 1$. Hence, in standard implementations of product spaces, a sphere $S_d$ is parametrized by a $(d + 1)$-dimensional vector [9]. However, this requires more coordinates to be stored for each spherical space. Hence, to make a fair comparison of all spaces, we use the hyperspherical coordinates for $S_d$:

$$\hat{M}_S(x) = \begin{pmatrix} \cos x_1 \cdot \cos x_2 \cdot \ldots \cdot \cos x_{d-1} \cdot \cos x_d \\ \cos x_1 \cdot \cos x_2 \cdot \ldots \cdot \cos x_{d-1} \cdot \sin x_d \\ \cos x_1 \cdot \cos x_2 \cdot \ldots \cdot \sin x_{d-1} \\ \ldots \\ \sin x_1 \end{pmatrix} \in S_d. \tag{5}$$

Now we are ready to define an overlapping space. Consider two vectors $x, y \in \mathbb{R}^d$. Let $p_1, \ldots, p_k$ denote some subsets of coordinates, i.e., $p_i \subset \{1, \ldots, d\}$. We assume that together these subsets cover all coordinates, i.e., $\cup_{i=1}^{k} p_i = \{1, \ldots, d\}$. By $x[p_i]$ we denote a subvector of $x$ induced by $p_i$. Let $D_i \in \{E, S, H\}$. We define $d_i(x, y) := d_{D_i}\big(M_{D_i}(x[p_i]), M_{D_i}(y[p_i])\big)$ and aggregate these distances with arbitrary positive weights $w_1 \ldots w_k \in \mathbb{R}_+$:

$$d_O^{l1}(x, y) = \sum_{i=1}^{k} w_i d_i(x, y), $$
$$d_O^{l2}(x, y) = \left( \sum_{i=1}^{k} w_i d_i^2(x, y) \right)^{1/2}. \tag{6}$$

**Definition 1.** $O_d = \{x \in \mathbb{R}^d\}$ *equipped with a distance* $d_O^{l1}$ *or* $d_O^{l2}$ *defined in* (6) *is called an* overlapping space. *This space is defined by* $p_i$, $D_i$, *and* $w_i$.

Note that it is sufficient to assume that spherical and hyperbolic spaces have curvatures 1 and $-1$, respectively, since changing the curvature is equivalent to changing scale, which is captured by $w_i$. The following statement follows from the definition above and the fact that $d_E$, $d_S$, and $d_H$ are distances (see Appendix C for the proof).

---

[4]We refer to Appendix D for additional simple illustrations of the proposed idea.

**Statement 1.** *If $\cup_{i=1}^k p_i = \{1, \ldots, d\}$ and $w_1 \ldots w_k \in \mathbb{R}_+$, then $d_O^{l1}$, $d_O^{l2}$ are distances on $\mathbb{R}^d \times \mathbb{R}^d$, i.e., they satisfy the metric axioms.*

It is easy to see that overlapping spaces generalize product spaces. Indeed, if we assume $p_i \cap p_j = \emptyset$ for all $i \neq j$, then an overlapping space reduces to a product space. However, the fact that we allow $p_i \cap p_j \neq \emptyset$ gives us a significantly larger expressive power for the same dimension $d$.

### 3.2 Generalization with WIPS: OS-Mixed measure

It is known that for ranking loss functions, methods based on weighted and standard dot products can have good performance [13]. Let us note that such similarity measures cannot be converted to a distance via a monotone transformation. In our experiments, we notice that in some cases, such non-metric methods can successfully be used for graph embeddings even with the distortion loss, when the model approximates metric distances.

To close the gap between metric and non-metric methods, we propose a generalization of the overlapping spaces that also includes the weighted inner product similarity (WIPS).

First, let us motivate our choice of WIPS. One may extend the list of base distance functions $\{d_E, d_S, d_H\}$ with a dissimilarity measure $d_{\text{dot}} = c - x^T y$ that is not a metric distance. By using such 'distance' for all possible subsets $p_i \in 2^{\{1 \ldots d\}}$ and applying $l1$-aggregation (6), we get WIPS measure $d_W = \tilde{c} - \sum_{i=1}^d \tilde{w}_i x_i y_i$, where $\tilde{c}$ and $\tilde{w}_i$ are trainable values.

Thus, instead of using an extended set of base distances $\{d_E, d_S, d_H, d_{\text{dot}}\}$ together, as shown in equation (6), we suggest to simply use $d_{OS-Mixed}(x, y) = d_O(x, y) + d_W(x, y)$ where $d_O$ is a metric overlapping space. We show that this design gives excellent results for both distortion and ranking objectives in the graph reconstruction task. We further refer to this approach as *OS-Mixed*.

## 4 Optimization in overlapping spaces

### 4.1 Binary tree signature

Overlapping spaces defined in Section 3 are flexible and allow capturing various geometries. However, similarly to product spaces, they need a signature ($p_i$ and $D_i$) to be chosen in advance. This section describes a particular signature, which is still flexible, does not suffer from the brute-force problem, and shows promising empirical results on the datasets we experimented with.

Let $t \geq 0$ denote the depth (complexity) of the signature for a $d$-dimensional embedding. Each layer $l$, $0 \leq l \leq t$, of the signature consists of $2^l$ subsets of coordinates: $p_i^l = \left\{ \left\lceil d(i-1)/2^l \right\rceil + 1, \ldots, \left\lceil di/2^l \right\rceil \right\}$, $1 \leq i \leq 2^l$. Each $p_i^l$ is associated with Euclidean, spherical, and hyperbolic spaces simultaneously. The corresponding weights are denoted by $w_i^{l,E}, w_i^{l,S}, w_i^{l,H}$. Then, the distance is computed according to (6). See Figure 1 for an illustration of the procedure for $d = 10$ and $t = 1$.

Informally, assume that we have two vectors $x, y \in \mathbb{R}^d$. To compute the distance between them in the proposed overlapping space, we first compute Euclidean, spherical, and hyperbolic distances between $x$ and $y$. Then, we split the coordinates into two halves, and for each half, we also compute all three distances, and so on. Finally, all the obtained distances are averaged with the weight coefficient according to (6). Note that we have $3(2^{t+1} - 1)$ different weights in our structure in general, but with $l2$-aggregation this value may be reduced to $2(2^{t+1} - 1) + 2^t$ since for the Euclidean space, the distances between subvectors at the upper layers can be split into terms corresponding to smaller subvectors, so we essentially need only the last layer with $2^t$ terms.

Recall that in a product space, the weights correspond to curvatures of the combined hyperbolic and spherical spaces. In our case, they also play another important role: the weights allow us to balance between different spaces. Indeed, for each subset of coordinates, we simultaneously compute the distance between the points assuming each of the combined spaces. Varying the weights, we can increase or decrease the contribution of a particular space to the distance. As a result, our signature allows us to learn the optimal combination, which does not have to be a product space since all weights can be non-zero. Interestingly, when we analyzed the optimized weights, we observed that

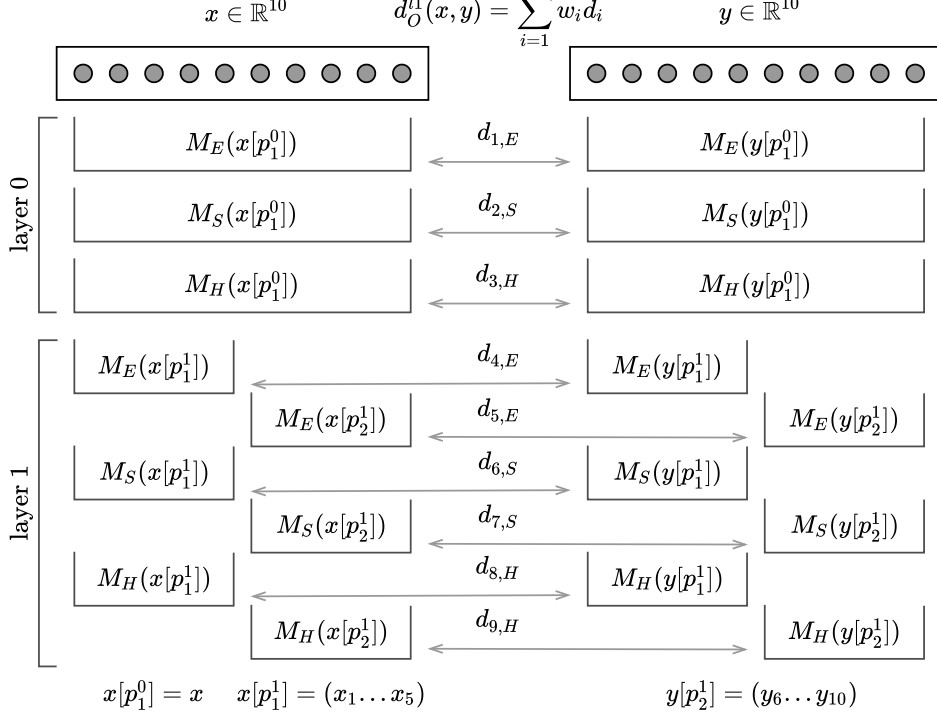

Figure 1: Overlapping space with $d = 10$, $t = 1$, and $l1$ (sum) aggregation

often some of them are close to zero. Thus, unnecessary weights can be detected and removed after the training. See Appendix B.5 for more details.

Note that the procedure described in this section naturally extends to OS-Mixed by adding the corresponding 'distance' to Euclidean, hyperbolic, and spherical as described in Section 3.2.

## 4.2 Optimization

In this section, we describe how we embed into the overlapping space. Although Riemann-SGD (see Section 2.3) is a good solution from the theoretical point of view, in practice, due to errors in storing and processing real numbers, it may cause some problems. Indeed, a point that we assume to lie on a surface (sphere or hyperboloid) does not numerically lie on it usually. Due to the accumulation of numerical errors, with each iteration of RSGD, a point may move away from the surface. Therefore, in practice, after each step, all embeddings are explicitly projected onto the surface, which may slow down the algorithm. Moreover, RSGD is not applicable if one needs to process the output of a neural network, which cannot be required to belong to a given surface (e.g., to satisfy $\langle x, x \rangle_h = 1 \Leftrightarrow x \in H_d$). As a result, before finding the hyperbolic distance between two outputs of a neural network in the Siamese [3] setup, one first needs to map them to a hyperboloid.

Instead of RSGD, we store the embedding vectors in Euclidean space and calculate distances between them using the mappings (4) to the corresponding surfaces. Thus, we can evaluate the distances between the outputs of neural networks and also use conventional optimizers. To optimize embeddings, we first map Euclidean vectors into the corresponding spaces, calculate distances and loss function, and then backpropagate through the projection functions. To improve the convergence, we use Adam [14] instead of the standard SGD. Similar technique using SGD with momentum is used in [15]. Applying this to product spaces, we achieve the results similar to the original paper [9] (see Table 7 in Appendix), where RSGD was used with the learning rate brute-forcing, custom learning rate for curvature coefficients, and other tricks.

Table 1: Datasets for graph reconstruction

|  | USCA312 | CS PhDs | Power | Facebook | WLA6 | EuCore |
|---|---|---|---|---|---|---|
| Nodes | 312 | 1025 | 4941 | 4039 | 3227 | 986 |
| Edges | 48516 (weighted) | 1043 | 6594 | 88234 | 3604 | 16687 |

Table 2: Distortion graph reconstruction, top results are highlighted, top metric results are underlined

| Signature | USCA312 | CS PhDs | Power | Facebook | WLA6 | EuCore |
|---|---|---|---|---|---|---|
| $E_{10}$ | **0.00318** | 0.0475 | 0.0408 | 0.0487 | 0.0530 | 0.1242 |
| $H_{10}$ | 0.01104 | 0.0443 | 0.0348 | 0.0483 | 0.0279 | 0.1144 |
| $S_{10}$ | 0.01065 | 0.0519 | 0.0453 | 0.0561 | 0.0608 | 0.1260 |
| $H_5^2 \equiv H_5 \times H_5$ | 0.00573 | 0.0345 | 0.0255 | 0.0372 | 0.0279 | 0.1106 |
| $S_5^2 \equiv S_5 \times S_5$ | 0.00700 | 0.0501 | 0.0438 | 0.0552 | 0.0584 | 0.1251 |
| $H_5 \times S_5$ | 0.00541 | 0.0341 | 0.0254 | 0.0346 | 0.0310 | 0.1195 |
| $H_2^5$ | 0.00592 | 0.0344 | 0.0273 | 0.0439 | 0.0356 | 0.1163 |
| $S_2^5$ | 0.00604 | 0.0464 | 0.0416 | 0.0512 | 0.0543 | 0.1244 |
| $H_2^2 \times E_2 \times S_2^2$ | 0.00537 | 0.0344 | 0.0302 | 0.0406 | 0.0437 | 0.1193 |
| $O_{l1}, t = 0$ | **0.00324** | 0.0368 | 0.0281 | 0.0458 | 0.0286 | 0.1141 |
| $O_{l1}, t = 1$ | 0.00325 | **0.0300** | 0.0231 | 0.0371 | 0.0272 | 0.1117 |
| $O_{l2}, t = 1$ | 0.00530 | 0.0328 | **0.0246** | 0.0324 | 0.0278 | 0.1127 |
| $c - \mathrm{dot}$ | 0.04005 | 0.0412 | 0.0461 | 0.0236 | 0.0296 | 0.1085 |
| $c - \mathrm{wips}$ | 0.06468 | 0.0358 | 0.0442 | 0.0161 | **0.0238** | 0.1016 |
| $ce^{-\mathrm{dot}}$ | 0.08142 | 0.0424 | 0.0505 | 0.0192 | 0.0270 | 0.1048 |
| $O_{mix-l1}, t = 1$ | 0.00277 | 0.0243 | 0.0235 | **0.0172** | 0.0187 | 0.1026 |
| $O_{mix-l2}, t = 1$ | 0.00464 | 0.0220 | 0.0258 | 0.0163 | 0.0198 | **0.1028** |

## 5 Experiments

### 5.1 Compared spaces

In this section, we provide a thorough analysis to compare all metric spaces discussed in the paper, including product spaces with all signatures from [9] and the proposed overlapping space. For the non-metric dissimilarity functions, we consider $d(x, y) = c - x^T y$, $d(x, y) = c - \sum w_i x_i y_i$ (WIPS), $d(x, y) = c \exp(-x^T y)$ with trainable parameters $c, w_i \in \mathbb{R}$, and the proposed OS-Mixed measure. We add non-metric measures even to the distortion setup to see whether they are able to approximate graph distances. Similarly to [9], we fix the dimension $d = 10$. However, for a fair comparison, we fix the number of *stored values* for each embedding and use the hyperspherical parametrization (5) instead of storing $d + 1$ coordinates.[5] The training details are given in Appendix A. The code of our experiments is available.[6]

### 5.2 Graph reconstruction

**Graph datasets** We use the following graph datasets: the USCA312 dataset of distances between North American cities [4] (weighted complete graph), a graph of computer science Ph.D. advisor-advisee relationships [1], a power grid distribution network with backbone structure [28], a dense social network from Facebook [17], and EuCore dataset generated using email data from a large European research institution [16]. We also collected a new dataset by launching the breadth-first search on the Wikipedia category graph, starting from the "Linear Algebra" category with search depth limited to 6. Further, we refer to this dataset as WLA6; more details are given in Appendix A.2. This graph is very close to being a tree, although it has some cycles. We expect the hyperbolic space to give a significant profit for this graph, and we observe that product spaces give almost no additional advantage. The purpose of using this additional dataset is to evaluate overlapping spaces on a dataset where product spaces do not provide quality gains. Table 1 lists the properties of all considered datasets.

---

[5]In Appendix B.2, we evaluate spherical spaces without this modification to compare with [9].

[6]https://github.com/shevkunov/overlapping-spaces-for-compact-graph-representations

Table 3: mAP graph reconstruction, top results are highlighted, top metric results are underlined

| Signature | USCA312 | CS PhDs | Power | Facebook | WLA6 | EuCore |
|---|---|---|---|---|---|---|
| $E_{10}$ | 0.9290 | 0.9487 | 0.9380 | 0.7876 | 0.7199 | 0.6108 |
| $H_{10}$ | 0.9173 | 0.9399 | 0.9385 | 0.7997 | 0.9617 | 0.6670 |
| $S_{10}$ | 0.9183 | 0.9519 | 0.9445 | 0.7768 | 0.7289 | 0.6037 |
| $H_5^2$ | 0.9247 | 0.9481 | 0.9415 | 0.8084 | 0.9682 | 0.6783 |
| $S_5^2$ | 0.9316 | 0.9600 | 0.9482 | 0.7790 | 0.7307 | 0.6116 |
| $H_5 \times S_5$ | 0.9397 | 0.9538 | 0.9505 | 0.7947 | 0.9751 | 0.6847 |
| $H_2^5$ | 0.9364 | 0.9671 | 0.9508 | 0.7979 | 0.8597 | 0.6611 |
| $S_2^5$ | 0.9439 | 0.9656 | 0.9511 | 0.7800 | 0.7358 | 0.6169 |
| $H_2^2 \times E_2 \times S_2^2$ | 0.9519 | 0.9638 | 0.9507 | 0.7873 | 0.7794 | 0.6492 |
| $O_{l1}, t = 0$ | **0.9538** | 0.9879 | 0.9728 | 0.8093 | 0.6759 | 0.6580 |
| $O_{l1}, t = 1$ | **0.9522** | **0.9904** | 0.9762 | 0.8185 | 0.9598 | 0.6691 |
| $O_{l2}, t = 1$ | **0.9522** | **0.9938** | 0.9907 | 0.8326 | 0.9694 | 0.7078 |
| $c - \mathrm{dot}$ | **1** | **1** | 0.9983 | 0.8745 | 0.9990 | 0.7409 |
| $c - \mathrm{wips}$ | **1** | **1** | **1** | 0.8704 | **1** | **0.7742** |
| $O_{mix-l1}, t = 1$ | **1** | **1** | 0.9994 | 0.8806 | 0.9997 | 0.7860 |
| $O_{mix-l2}, t = 1$ | **1** | **1** | **1** | 0.9021 | **1** | 0.8405 |

**Distortion loss** We start with the standard graph reconstruction task with distortion loss (1). The goal is to embed all nodes of a given graph into a $d$-dimensional space approximating the pairwise graph distances between them. In this setup, all models are trained to minimize distortion (1), the results are shown in Table 2. It can be seen that the overlapping spaces outperform other metric spaces, and the best overlapping space (among considered) is the one with $l1$ aggregation and complexity $t = 1$.[7] Interestingly, the performance of such overlapping space is often better than the *best* considered product space.

We note that standard non-metric distance functions show highly unstable results for this task: for the USCA312 dataset, the obtained distortion is orders of magnitude worse than the best one. However, on some datasets (Facebook and WLA6), the performance is quite good, and for Facebook, these simple non-metric similarities have much better performance than all metric solutions. Thus, we conclude that such functions are worth trying for graph reconstruction with the distortion loss, but their performance is unstable. In contrast, the overlapping spaces show good and stable results on all datasets, and the proposed OS-Mixed modification (see Section 3.2) outperforms all other approaches.

**Ranking loss** As discussed in Section 2.1, in many practical applications, only the order of the nearest neighbors matters. In this case, it is more reasonable to use mAP (2). In previous work [9], mAP was also reported, but the models were trained to minimize distortion. In our experiments, we observe that distortion optimization weakly correlates with mAP optimization. Hence, we minimize the proxy-loss defined in equation (3). The results are shown in Table 3, and the obtained values for mAP are indeed much better than the ones obtained with distortion optimization [9], i.e., it is essential to use an appropriate loss function. According to Table 3, among the metric spaces, the best results are achieved with the overlapping spaces (especially for $l2$-aggregation with $t = 1$). Importantly, in contrast to distortion loss, ranking based on the dot-product outperforms all metric spaces. However, using the OS-Mixed measure allows us to improve these results even further.

## 5.3 Information retrieval

From a practical perspective, it is also important to analyze whether an embedding can generalize to unseen examples. For instance, an embedding can be made via a neural network based on objects' characteristics, such as text descriptions or images. This section analyzes whether it is reasonable to use complex geometries, including product spaces and overlapping spaces, in such a scenario.

---

[7]In Appendix B.4 we also analyze $t > 1$ and show that performance improves as $t$ increases.

[8]The gap between $E_{256}$ and $H_{256}$ may seem suspicious, but in Table 5 of [9] a similar pattern is observed.

Table 4: DSSM embeddings, top three results are highlighted

| Signature | Test mAP |
|---|---|
| $E_{10}$ | 0.4459 |
| $H_{10}$ | 0.4047 |
| $S_{10}$ | 0.4364 |
| $H_5^2$ | 0.4492 |
| $S_5^2$ | **0.4573** |
| $H_5 \times S_5$ | 0.3295 |
| $H_2^5$ | 0.3681 |
| $S_2^5$ | **0.4616** |
| $H_2^2 \times E_2 \times S_2^2$ | 0.3526 |
| $c - \mathrm{dot}$ | 0.4194 |
| $O_{l1}, t = 0$ | **0.4562** |
| $O_{l1}, t = 1$ | 0.4498 |
| $O_{l2}, t = 1$ | 0.4456 |
| $O_{mix-l1}, t = 1$ | 0.4447 |
| $O_{mix-l2}, t = 1$ | 0.4483 |

| Signature | Test mAP |
|---|---|
| $E_{256}$ | **0.717** |
| $H_{256}$ | 0.412 [8] |
| $S_{255}$ | 0.588 |
| $H_{128}^2$ | 0.547 |
| $S_{127}^2$ | 0.662 |
| $H_{128} \times S_{127}$ | 0.501 |
| $H_{51}^4 \times H_{52}$ | 0.621 |
| $S_{50}^4 \times S_{51}$ | **0.701** |
| $c - \mathrm{dot}$ | **0.738** |
| $O_{l1}, t = 0$ | 0.677 |
| $O_{l1}, t = 1$ | 0.662 |
| $O_{mix-l1}, t = 1$ | 0.663 |
| $O_{mix-l2}, t = 1$ | 0.655 |

For this purpose, we trained a classic DSSM model[9] [11] on a private Wikipedia search dataset consisting of 491044 pairs (search query, relevant page), examples are given in Table 11 in Appendix. All queries are divided into train, validation, and test sets, and for each signature, the optimal iteration was selected on the validation set. Table 4 compares all models for two embedding sizes. For short embeddings, we see that a product space based on spherical geometry is useful, and overlapping spaces have comparable quality. However, for large "industrial size" dimensions, the best results are achieved with the standard dot product, questioning the utility of complex geometries in the case of large dimensions.

Note that in DSSM-like models, the most time-consuming task is model training. Hence, training multiple models for choosing the best configuration can be infeasible. Therefore, for small dimensions, overlapping spaces can be preferable over product spaces since they are universal and do not require parameter tuning. Moreover, calculating element embeddings is usually more time-consuming than calculating distances. Hence, even though calculating distances in the overlapping space has larger complexity than in simpler spaces, it does not have a noticeable effect in real applications.

## 5.4 Synthetic bipartite graph reconstruction

Let us additionally illustrate that some graph structures are hard to embed into (considered) metric spaces. Our intuition is that the dot product is suitable for modelling *popularity* of items since it can be naturally reflected using vector norms. Thus, we consider a situation when there are a few objects that are more popular than the other ones. To model this, we consider a synthetic bipartite graph with two sets of sizes 20 and 700 with 5% edge probability (isolated nodes were removed, and the remaining graph is connected). Clearly, in the obtained graph, there are a few popular nodes and many nodes of small degrees. Figure 2 visualizes the obtained graph. Table 5 compares the performance of the best metric space with the dot-product performance. As we can see, this experiment confirms our intuition that some specific graphs are hard to embed into metric spaces, even with distortion loss. We also see that our OS-Mixed approach gives the best result with a margin. Thus, this experiment additionally confirms the universality of the proposed approach. We also observe that the optimization of WIPS is highly unstable on this dataset, see Table 6 for details.

---

[9]We changed dense layers sizes in order to achieve the required embedding length and used more complex text tokenization with char bigrams, trigrams, and words, instead of just char trigrams.

Table 6: WIPS distortion (5 restarts; best learning rate)

|  | avg. | worst | best | std |
|---|---|---|---|---|
| $c - \text{wips}$ | 0.092 | 0.100 | 0.078 | 0.0078 |
| $O_{mix-l2}, t = 1$ | 0.071 | 0.074 | 0.069 | 0.0018 |

Table 5: Bipartite graph reconstruction

|  | mAP | distortion |
|---|---|---|
| $E_{10}$ | 0.777 | 0.094 |
| $H_{10}$ | 0.794 | 0.095 |
| $S_{10}$ | 0.796 | 0.096 |
| $H_5^2$ | 0.799 | 0.090 |
| $S_5^2$ | 0.796 | 0.094 |
| $H_5 \times S_5$ | 0.798 | 0.090 |
| $H_2^5$ | 0.761 | 0.086 |
| $S_2^5$ | 0.773 | 0.092 |
| $H_2^2 \times E_2 \times S_2^2$ | 0.796 | 0.089 |
| $O_{l1}, t = 0$ | 0.824 | 0.094 |
| $O_{l1}, t = 1$ | 0.803 | 0.082 |
| $O_{l2}, t = 1$ | 0.814 | 0.092 |
| best metric space | 0.824 | 0.082 |
| $c - \text{dot}$ | 0.863 | 0.079 |
| $c - \text{wips}$ | 1 | 0.091 |
| $O_{mix-l1}, t = 1$ | 0.986 | 0.083 |
| $O_{mix-l2}, t = 1$ | 1 | 0.070 |

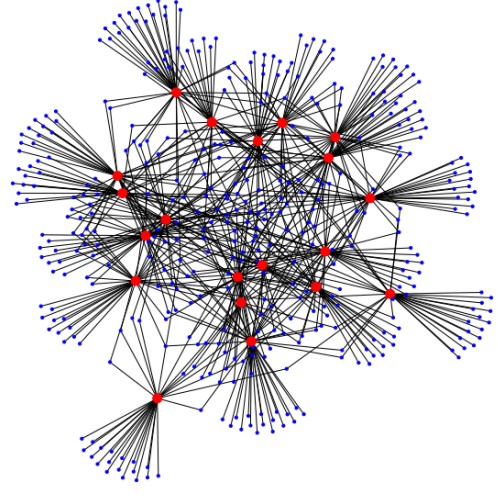

Figure 2: Graph visualization. Red (big) nodes belong to the smaller part.

# 6 Conclusion

This paper proposed the new concept of overlapping spaces that do not require signature brute-forcing and have better or comparable performance relative to the best product space in the graph reconstruction task. Improvements are observed for both global distortion and local mAP loss functions. An important advantage of our method is that it allows us to easily incorporate new distances or similarities as building blocks. The obtained overlapping-mixed non-metric measure achieves the best results for both distortion and mAP. We also evaluated the proposed overlapping spaces in the DSSM setup, and in the case of short embeddings, product space gives a better result than standard spaces, and the OS is comparable to it. In the case of long embeddings, no profit from complex spaces was found.

## Acknowledgments and Disclosure of Funding

The authors would like to thank Egor Samosvat for fruitful discussions.

The work of Liudmila Prokhorenkova is partially supported by the Ministry of Education and Science of the Russian Federation in the framework of MegaGrant 075-15-2019-1926 and by the Russian President grant supporting leading scientific schools of the Russian Federation NSh2540.2020.1. Besides that, the funding was provided exclusively by the companies listed in the authors' affiliation.

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
