# A Experimental setup

## A.1 Training details

All models discussed in Section 5.2 were trained with 2000 iterations. If more than one learning rate was used for a certain dataset (due to problems with the convergence of individual models), all the spaces were evaluated for all learning rates, and the best result was reported for each space. For distortion, the learning rate was 0.1 for all datasets except USCA312 (Cities), where we had 0.1 and 0.01. For mAP, the learning rate 0.1 was used for all datasets except USCA312 and CSPhDs, where we had 0.01 and 0.05 for both datasets.

For the experiments in Section 5.3 , we used 5000 iterations for short embeddings and 1000 for long ones (long embeddings converged faster). Hard-negative mining was not used for DSSM training. Instead, large batches of 4096 random training examples (almost 1% of the entire dataset) were used. During the learning process, only the training queries and documents were used. For evaluation, the nearest website was searched among all the documents. The training part was 90% of the dataset, and the quality discrepancy between validation and test sets was quite small. Data samples are given in the table 11.

For the synthetic experiment in Section 5.4 , for all spaces, the learning rates 0.1, 0.05, 0.01, 0.001 were used, and the best result was selected. We had 2000 and 1000 iterations for distortion and mAP, respectively.

## A.2 WLA6 dataset details

As described in the main text, this dataset is obtained by running the breadth-first search algorithm on the category graph of the English-language Wikipedia (`https://en.wikipedia.org/wiki/Special:CategoryTree`), starting from the vertex (category) "Linear algebra" and limited to the depth 6 (Wikipedia Linear Algebra 6). We provide this graph along with the texts (names) of the vertices (categories). The resulting graph is very close to being a tree, although there are some cycles. Predictably, hyperbolic space gives a significant profit for this graph, while using product spaces gives almost no additional advantage. The purpose of using this dataset is to check our conclusions on data other than those used in [9] and to evaluate overlapping spaces on a dataset where product spaces do not provide quality gains.

# B Additional experimental results

## B.1 Our implementation of product spaces vs original one

Table 7 compares our implementation with the results reported in [9]. It should be noted that we have significantly different algorithms with differing numbers of iterations.

The optimal values of distortion obtained with our algorithm (except for the USCA312 dataset) are comparable and usually better than those reported in [9]. On USCA312, the obtained distortion is orders of magnitude better, which can be caused by the proper choice of the learning rate (in our experiments on this dataset, this choice significantly affected the results). These results indicate that our solution is a good starting point to compare different spaces and similarities.

For mAP, we optimize the proxy-loss, in contrast to the canonical implementation, where both metrics were specified for models trained with distortion. Clearly, the results are more stable for our approach: we do not have such a large spread of values for different spaces. We noticed that directly optimizing ranking losses leads to significant improvements.

## B.2 Parametrization of spherical space

In Tables 2 and 3 of the main text, we used hyperspherical parameterization of spherical subspaces in product spaces since we fixed the number of stored values for each space. Here, in Tables 8 and 9, we present the extended results, where we fix the mathematical dimension of product spaces and use $d + 1$ parameters and simple mappings from Section 3, equation (4),

Table 7: Graph reconstruction: original product spaces vs our implementation

|  | USCA312 | | CS PhDs | | Power | | Facebook | |
|---|---|---|---|---|---|---|---|---|
|  | Canon. | Our | Canon. | Our | Canon. | Our | Canon. | Our |
| Distortion | | | | | | | | |
| $E_{10}$ | 0.0735 | 0.0032 | 0.0543 | 0.0475 | 0.0917 | 0.0408 | 0.0653 | 0.0487 |
| $H_{10}$ | 0.0932 | 0.0111 | 0.0502 | 0.0443 | 0.0388 | 0.0348 | 0.0596 | 0.0483 |
| $S_{10}$ | 0.0598 | 0.0095 | 0.0569 | 0.0503 | 0.0500 | 0.0450 | 0.0661 | 0.0540 |
| $H_5 \times H_5$ | 0.0756 | 0.0057 | 0.0382 | 0.0345 | 0.0365 | 0.0255 | 0.0430 | 0.0372 |
| $S_5 \times S_5$ | 0.0593 | 0.0079 | 0.0579 | 0.0492 | 0.0471 | 0.0433 | 0.0658 | 0.0511 |
| $H_5 \times S_5$ | 0.0622 | 0.0068 | 0.0509 | 0.0337 | 0.0323 | 0.0249 | 0.0402 | 0.0318 |
| $H_2^5$ | 0.0687 | 0.0059 | 0.0357 | 0.0344 | 0.0396 | 0.0273 | 0.0525 | 0.0439 |
| $S_2^5$ | 0.0638 | 0.0072 | 0.0570 | 0.0460 | 0.0483 | 0.0418 | 0.0631 | 0.0489 |
| $H_2^2 \times E_2 \times S_2^2$ | 0.0765 | 0.0044 | 0.0391 | 0.0345 | 0.0380 | 0.0299 | 0.0474 | 0.0406 |
| mAP | | | | | | | | |
| $E_{10}$ |  | 0.9290 | 0.8691 | 0.9487 | 0.8860 | 0.9380 | 0.5801 | 0.7876 |
| $H_{10}$ |  | 0.9173 | 0.9310 | 0.9399 | 0.8442 | 0.9385 | 0.7824 | 0.7997 |
| $S_{10}$, |  | 0.9254 | 0.8329 | 0.9578 | 0.7952 | 0.9436 | 0.5562 | 0.7868 |
| $H_5 \times H_5$ |  | 0.9247 | 0.9628 | 0.9481 | 0.8605 | 0.9415 | 0.7742 | 0.8084 |
| $S_5 \times S_5$ |  | 0.9231 | 0.7940 | 0.9662 | 0.8059 | 0.9466 | 0.5728 | 0.7891 |
| $H_5 \times S_5$ |  | 0.9316 | 0.9141 | 0.9654 | 0.8850 | 0.9467 | 0.7414 | 0.8087 |
| $H_2^5$ |  | 0.9364 | 0.9694 | 0.9671 | 0.8739 | 0.9508 | 0.7519 | 0.7979 |
| $S_2^5$ |  | 0.9281 | 0.8334 | 0.9714 | 0.8818 | 0.9521 | 0.5808 | 0.7915 |
| $H_2^2 \times E_2 \times S_2^2$ |  | 0.9391 | 0.8672 | 0.9611 | 0.8152 | 0.9486 | 0.5951 | 0.7970 |

as done in [9]. We can see that our implementation gives results comparable to the original ones in distortion setup and significantly better for mAP, which is associated with using the proxy-loss instead of distortion.

Table 8: Graph reconstruction with distortion loss, top results are highlighted, metrics only

| Signature | USCA312 | CS PhDs | Power | Facebook | WLA6 |
|---|---|---|---|---|---|
| $E_{10}$ | **0.00318** | 0.0475 | 0.0408 | 0.0487 | 0.0530 |
| $H_{10}$ | 0.01114 | 0.0443 | 0.0348 | 0.0483 | **0.0279** |
| $S_{10}$ | 0.00951 | 0.0503 | 0.0450 | 0.0540 | 0.0589 |
| $H_5^2 \equiv H_5 \times H_5$ | 0.00573 | 0.0345 | 0.0255 | 0.0372 | **0.0279** |
| $S_5 \times S_5 \equiv S_5^2$ | 0.00792 | 0.0492 | 0.0433 | 0.0511 | 0.0585 |
| $H_5 \times S_5$ | 0.00681 | **0.0337** | **0.0249** | **0.0318** | 0.0296 |
| $H_2^5$ | 0.00592 | 0.0344 | 0.0273 | 0.0439 | 0.0356 |
| $S_2^5$ | 0.00720 | 0.0460 | 0.0418 | 0.0489 | 0.0549 |
| $H_2^2 \times E_2 \times S_2^2$ | 0.00436 | 0.0345 | 0.0299 | 0.0406 | 0.0405 |
| $O_{l1}, t = 0$ | **0.00356** | 0.0368 | 0.0281 | 0.0458 | 0.0286 |
| $O_{l1}, t = 1$ | **0.00330** | **0.0300** | **0.0231** | **0.0371** | **0.0272** |
| $O_{l2}, t = 1$ | 0.00530 | **0.0328** | **0.0246** | **0.0324** | **0.0278** |

Table 10: Comparison of proxy-losses, mAP

| | USCA312 | | | CS PhD | | |
|---|---|---|---|---|---|---|
| $P \sim$ | $e^{-d}$ | $e^{1/d}$ | $1/d$ | $e^{-d}$ | $e^{1/d}$ | $1/d$ |
| $E_{10}$ | 0.929 | 0.911 | 0.899 | 0.949 | 0.956 | 0.831 |
| $H_{10}$ | 0.917 | 0.807 | 0.885 | 0.940 | 0.749 | 0.764 |
| $S_{10}$ | 0.925 | 0.797 | 0.838 | 0.958 | 0.572 | 0.689 |
| $H_5^2$ | 0.925 | 0.890 | 0.883 | 0.948 | 0.976 | 0.723 |
| $S_5^2$ | 0.923 | 0.802 | 0.858 | 0.966 | 0.748 | 0.775 |
| $H_5 \times S_5$ | 0.932 | 0.838 | 0.865 | 0.965 | 0.804 | 0.721 |
| $H_2^5$ | 0.936 | 0.896 | 0.903 | 0.967 | 0.998 | 0.823 |
| $S_2^5$ | 0.928 | 0.856 | 0.871 | 0.971 | 0.876 | 0.881 |
| $H_2^2 \times E_2 \times S_2^2$ | 0.939 | 0.872 | 0.865 | 0.961 | 0.884 | 0.689 |
| $O_{l1}, t=0$ | 0.952 | 0.933 | 0.872 | 0.988 | 0.961 | 0.762 |
| $O_{l1}, t=1$ | 0.952 | 0.947 | 0.877 | 0.990 | 0.963 | 0.815 |
| $O_{l2}, t=1$ | 0.952 | 0.939 | 0.880 | 0.994 | 0.979 | 0.810 |
| $c - \mathrm{dot}$ | 1 | 1 | 0.777 | 1 | 0.999 | 0.917 |

Table 9: Graph reconstruction with mAP ranking loss, top results are highlighted, metrics only

| Signature | USCA312 | CS PhDs | Power | Facebook | WLA6 |
|---|---|---|---|---|---|
| $E_{10}$ | 0.9290 | 0.9487 | 0.9380 | 0.7876 | 0.7199 |
| $H_{10}$ | 0.9173 | 0.9399 | 0.9385 | 0.7997 | 0.9617 |
| $S_{10}$ | 0.9254 | 0.9578 | 0.9436 | 0.7868 | 0.7287 |
| $H_5^2$ | 0.9247 | 0.9481 | 0.9415 | 0.8084 | **0.9682** |
| $S_5^2$ | 0.9231 | 0.9662 | 0.9466 | 0.7891 | 0.7353 |
| $H_5 \times S_5$ | 0.9316 | 0.9654 | 0.9467 | 0.8087 | **0.9779** |
| $H_2^5$ | 0.9364 | 0.9671 | 0.9508 | 0.7979 | 0.8597 |
| $S_2^5$ | 0.9281 | 0.9714 | 0.9521 | 0.7915 | 0.7346 |
| $H_2^2 \times E_2 \times S_2^2$ | 0.9391 | 0.9611 | 0.9486 | 0.7970 | 0.6796 |
| $O_{l1}, t=0$ | **0.9522** | **0.9879** | **0.9728** | **0.8093** | 0.6759 |
| $O_{l1}, t=1$ | **0.9522** | **0.9904** | **0.9762** | **0.8185** | 0.9598 |
| $O_{l2}, t=1$ | **0.9522** | **0.9938** | **0.9907** | **0.8326** | **0.9694** |

## B.3 Other ways of converting distances to probabilities

For the proxy-loss, we additionally experimented with other ways of converting distances to probabilities. Let us write $L_{proxy}$ in the general form:

$$L_{proxy} = - \sum_{(v,u) \in E} \log \mathrm{P}((v,u) \in E) = - \sum_{(v,u) \in E} \log \frac{t\big(d_U(f(v), f(u))\big)}{\sum\limits_{w \in V} t\big(d_U(f(v), f(w))\big)}, \qquad (7)$$

where $t(d)$ is a function that decreases with distance $d$. We compare the following alternatives for $t(d)$:

$$t_1(d) = \exp(-d), t_2(d) = \exp\left(\frac{1}{\min(d, d_0)}\right), t_3(d) = \frac{1}{\min(d, d_0)},$$

where $d_0$ is a small constant.

Recall that $t_1$ was used in the main text and it seems to be the most natural choice.[10] Table 10 compares the options and shows that the best results are indeed achieved with $t_1$.

---

[10]Note that this is the softmax over the inverted distances.

Table 11: Search query examples

| Query | Web site |
|---|---|
| Kris Wallace | en.wikipedia.org/wiki/Chris_Wallace |
| 1980: Mitsubishi produces one million cars...// | en.wikipedia.org/wiki/Mitsubishi_Motors |
| code napoleon | en.wikipedia.org/wiki/Napoleonic_Code |

Table 12: Distortion graph reconstruction for different overlapping spaces

| Signature | USCA312 | CS PhDs | Power | Facebook | WLA6 | EuCore |
|---|---|---|---|---|---|---|
| $O_{l1}, t=0$ | 0.00324 | 0.0368 | 0.0281 | 0.0458 | 0.0286 | 0.1141 |
| $O_{l1}, t=1$ | 0.00325 | 0.0300 | 0.0231 | 0.0371 | 0.0272 | 0.1117 |
| $O_{l1}, t=2$ | 0.00296 | 0.0335 | 0.0262 | 0.0309 | 0.0273 | 0.1114 |
| $O_{l1}, t=3$ | 0.00257 | 0.0273 | 0.0209 | 0.0313 | 0.0246 | 0.1098 |
| $O_{l2}, t=1$ | 0.00530 | 0.0328 | 0.0246 | 0.0324 | 0.0278 | 0.1127 |
| $O_{l2}, t=2$ | 0.00596 | 0.0303 | 0.0256 | 0.0312 | 0.0278 | 0.1117 |
| $O_{l2}, t=3$ | 0.00303 | 0.0343 | 0.0240 | 0.0302 | 0.0279 | 0.1119 |

## B.4 Analysis of depth in overlapping spaces

Distortion graph reconstruction results for all possible $t \le \log_2(d) = \log_2(10) \sim 3.3$ are provided in Table 12 for completeness. The results below confirm our hypothesis that the reconstruction distortion improves with increasing $t$.

## B.5 Analysis of learned weights

While analyzing the trained weights we have made several observations:

1. We see that OS does not learn a pure product space. In particular, on the CS PhDs dataset we get
$$d_{O_{l=1}, t=0} \propto 0.37 d_H + 0.63 d_S,$$
which is significantly better than both $d_S$ and $d_H$ separately.

2. If for $t = 0$ there is a space with a noticeably larger weight compared to the other ones, then the space of same type often makes the largest contribution for $t = 1$ too. For example, in USCA312,
$$d_{O_{l1}, t=0} \propto \mathbf{0.90} d_E + 0.05 d_H + 0.05 d_S,$$
and the weights of the Euclidean subdistances for $d_{O_{l1}, t=1}$ (normalized, $\sum w_i = 1$) are 0.6, 0.15, 0.1.

3. However, a space that is absent for $t = 0$ can appear for $t = 1$. For example, in the Power dataset,
$$d_{O_{l1}, t=0} \propto 0.37 d_H + 0.63 d_S,$$
$$d_{O_{l1}, t=1} \propto \mathbf{0.1} d_E(l_1^0, r_1^0) + 0.5 d_R(l_1^0, r_1^0) + 0.4 d_H(l_1^1, r_1^1),$$
where $l_1^0 = l[0..5], l_1^1 = l[6..10]$.

4. Finally, we noticed that almost always, more than half of the weights are near-zero, which allows one to remove unnecessary distances and improve efficiency.

## C  Proof of Statement 1

To prove that $d(x, y)$ is a metric distance, we need to show that it is symmetric, nonnegative, equals zero only when $x = y$, and satisfies the triangle inequality.

Consider an overlapping space:

$$d_O(l, r) = \mathrm{Agg}(d_{D_1}(\cdot, \cdot), \dots, d_{D_k}(\cdot, \cdot)),$$

where Agg is $l1$ or $l2$ aggregation and $d_{D_i}$ are base distances applied to subsets of coordinates.

Symmetry of $d_O$ follows from symmetry of base distances $d_{D_i}$. Obviously, we have $d_O(x,y) \geq 0$ and $d_O(x,x) = 0$. The inequality $d_O(x,y) > 0$ for $x \neq y$ follows from the fact hat we use specific non-trival mappings $M_{D_i}$ and assume that together subsets of coordinates $p_i$ cover all coordinates (i.e., $\cup_{i=1}^{k} p_i = \{1, \dots, d\}$).

Obviously, $l1$ aggregation (sum) preserves the triangle inequality. So, it remains to show this for $l2$. Assume that $d_1$ and $d_2$ satisfy the triangle inequality, nonnegative and let $d_{l2} = \sqrt{d_1^2 + d_2^2}$.

Let $c_1 := d_1(x,y)$, $c_2 := d_2(x,y)$, $a_1 = d_1(x,z)$, $a_2 = d_2(x,z)$, $b_1 = d_1(z,y)$, $b_2 = d_2(z,y)$.

We know that $c_1 \leq a_1 + b_1$ and $c_2 \leq a_2 + b_2$. Therefore,

$$c_1^2 + c_2^2 \leq a_1^2 + a_2^2 + b_1^2 + b_2^2 + 2a_1b_1 + 2a_2b_2 \,. \tag{8}$$

We need to show

$$\sqrt{c_1^2 + c_2^2} \leq \sqrt{a_1^2 + a_2^2} + \sqrt{b_1^2 + b_2^2} \,,$$

$$c_1^2 + c_2^2 \leq a_1^2 + a_2^2 + b_1^2 + b_2^2 + 2\sqrt{a_1^2 + a_2^2}\sqrt{b_1^2 + b_2^2} \,.$$

Taking into account Equation 8, it is sufficient to show

$$2a_1b_1 + 2a_2b_2 \leq 2\sqrt{a_1^2 + a_2^2}\sqrt{b_1^2 + b_2^2} \,,$$

$$a_1^2b_1^2 + a_2^2b_2^2 + 2a_1b_1a_2b_2 \leq (a_1^2 + a_2^2)(b_1^2 + b_2^2) \,,$$

$$2a_1b_1a_2b_2 \leq a_1^2b_2^2 + a_2^2b_1^2 \,,$$

which is true.

Finally, note that that for three base distances we have $d_{l2} = \sqrt{(\sqrt{d_1^2 + d_2^2})^2 + d_3^2} = \sqrt{d_1^2 + d_2^2 + d_3^2}$ and so we have proved the statement for an arbitrary number of terms.

## D  Additional illustrations for overlapping spaces

Figure 3 additionally illustrates the idea behind overlapping spaces. Namely, Figure 3(a) shows standard Euclidean distance evaluation between two vectors $l$ and $r$. As shown in Figure 3(b), we add a differentiable mapping $M_H : \mathbb{R}^{10} \to H_{10}$ to calculate the distance in the hyperbolic space (we may do the same for the spherical space). Applying several mappings to different parts of $l$ and $r$, we may get any product space as shown in Figure 3(c). The last step is to allow the subsets of coordinates to *overlap*, as shown in Figure 3(d), where the fifth coordinate is used simultaneously in two mappings. All such spaces with all possible intersections and base distances are called overlapping spaces.

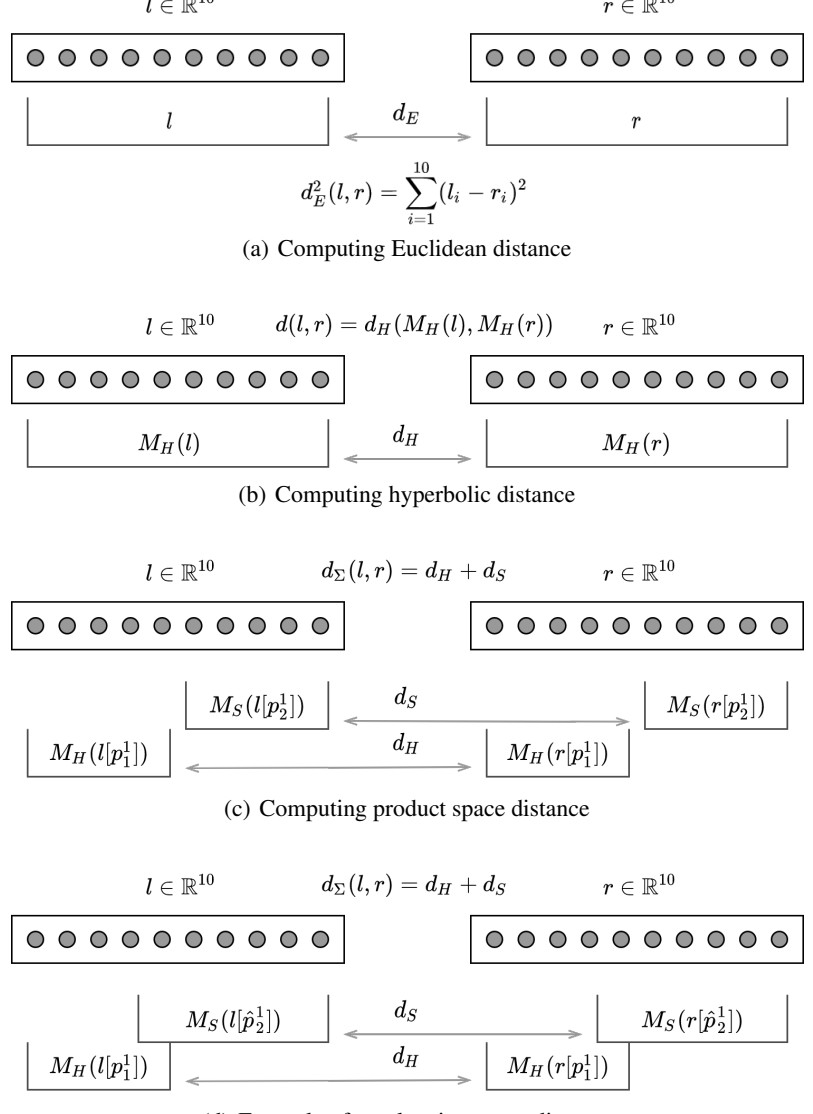

(a) Computing Euclidean distance

(b) Computing hyperbolic distance

(c) Computing product space distance

(d) Example of overlapping space distance

Figure 3: Illustrating overlapping space with $d = 10$ and $l1$ (sum) aggregation