# OpenReview forum: "Overlapping Spaces for Compact Graph Representations"
_NeurIPS.cc/2021/Conference — NeurIPS 2021 Poster_

### Official Review · Reviewer_ndZi · 2021-07-14

**Rating:** 7
**Confidence:** 4

**Summary:**

In this paper, the authors propose to extend product space for embedding structured data. In doing so, the author poses to allow each dimension to possibly belong to multiple signatures t. To optimize, the authors propose to use universal signatures and a regular Adam optimizer. Extensive experiments have been conducted to demonstrate the effectiveness of new embedded space.

**Limitations And Societal Impact:**

The authors adequately addressed the limitations and potential negative societal impact of their work.

**Main Review:**

In this paper, the authors propose to extend product space for embedding structured data, after identifying its issue that finding the best combination of signatures (a signature is a group of dimension associates with a distance and scale coefficient) is labor and resource-intensive. In doing so, the author poses to allow each dimension to possibly belong to multiple signatures t. To optimize, the authors propose to use universal signatures that effectively serve as aggregations of signatures, and to use regular Adam optimizer. Extensive experiments have been conducted to demonstrate the effectiveness of new embedded space.

Generally speaking, this paper is a nice extension over product space [1] which itself already shows improvement over traditional, single spaces for embedding structured data. Product space divides dimensions into multiple disjoint signatures, thus suffering from the need to manually select configuration of these signatures. By allowing each dimension to possibly multiple signatures. This paper addresses its issue but extends its expressiveness.

Technically this paper is well written and easy to follow. Details look sounding, expect for some parts of the optimization and experiments, which I have concerns (or confusions) as fellows:


- Regarding the universal signature. If I understand correctly, it is similar to using divide and conquer to aggregate in order to reduce computation cost from exponential $O(2^d)$ to linear $O(d)$ This computational feasibility would be important as this allows scalability, but I'm not sure
  - (a) why in the paper, only $t=0$ and $t=1$ is considered, instead of larger $t$. Note that $t \leq log_2 (n)$ are all valid choices.
  - (b) why the proposed universal signature can serve as a (possibly "approximated") replacement for a combination of all possible $2^d-1$ signatures. There seems to be no proof or experiments addressing  these issues.

- Regarding experiments: Only $d=10$, which is the same for [1] that this paper extends, is used for this work. What is the rationale behind such a choice?

- Relation with [1]. As an extension, this paper naturally borrows concepts and notations from [1]. However, as this paper itself should be a self-contained work, It's better to add more discuss  --- currently the notations are sometimes left unexplained and readers may need to refer to [1] to get the whole picture.

To summarize, this paper focuses on an extension over [1]. The improvements may be somewhat incremental but show us the possibility of dimensions being used for multiple spaces/metrics in embedding structured data. I would recommend it more strongly if my concerns are resolved.

References:
- [1] Gu et al. Learning mixed-curvature representations in product spaces. ICLR 2019. https://openreview.net/forum?id=HJxeWnCcF7



**After authors’ responses and discussion**: Some of my concerns are addressed and I increase my score from 6 to 7.


**Time Spent Reviewing:**

4 hours

---

> ### Author Response · Authors · 2021-08-10
> **Authors' response**
>
> Thank you very much for the feedback!
>
> - **Why in the paper, only t=0 and t=1 is considered**
>
>     In the paper, we consider the two simplest versions with t = 0 and t = 1, and for these models we already obtained promising results. Since all spaces covered by the case $t = 1$ are also covered by $t > 1$, we expect that $t > 1$ will give the quality no worse than that of $t = 1$. To verify this, we conducted additional experiments for distortion with $t = 2$ and found that the quality is indeed comparable or better, which confirms our assumptions:
>
>     | Signature | UCSA312 | CS PhDs | Power | Facebook | WLA6 | EuCore |
> | ------------- | ------------- | ------------- | -------- | ------------- | -------- | ----------- |
> |$O_{l1}, t=0$ | 0.00324 | 0.0368 | 0.0281 | 0.0458 | 0.0286 | 0.1141 |
> |$O_{l1}, t=1$ | 0.00325 | 0.0300 | 0.0231 | 0.0371 | 0.0272 | 0.1117 |
> |$O_{l1}, t=2$ | 0.00296 | 0.0335 | 0.0262 | 0.0309 | 0.0273 | 0.1114 |
> |$O_{l2}, t=1$ | 0.00530 | 0.0328 | 0.0246 | 0.0324 | 0.0278 | 0.1127 |
> |$O_{l2}, t=2$ | 0.00596 | 0.0303 | 0.0256 | 0.0307 | 0.0278 | 0.1117 |
>
>
> - **Why the proposed universal signature can serve as a (possibly "approximated") replacement
> for a combination of all possible $2^d-1$ signatures?**
>
>     In this section, when mentioning universality, we mean the fact that this approach works equally well on all the considered datasets and allows one to effectively use the overlapping spaces in practice without having to select a particular signature for each individual dataset as it happens in product space setup. The question of how well it works among all possible overlapping spaces is open and has not been considered in this paper. Thus, we agree that the term "universal" can be misleading here.
>
> - **Only $d= 10$ ... is used for this work. What is the rationale behind such a choice?**
>
>     We choose the value $d = 10$ to obtain results comparable with previous research [9]. In the original article, this dimension was chosen because for the considered graph datasets larger dimension is not required. Note that the difference between spaces is usually more significant in smaller dimensions since almost anything can be embedded in large ones.
>
> - **As an extension, this paper naturally borrows concepts and notations from [1]. However, as this paper itself should be a self-contained work, It's better to add more discuss - currently the notations are sometimes left unexplained and readers may need to refer to [1] to get the whole picture.**
>
>     We will proofread and extend this part to make sure that the text is self-contained.

---

> > ### Comment · Reviewer_ndZi · 2021-08-26
> > **Response to Authors**
> >
> >
> > I thank the authors for their effort in responding to my reviews.
> >
> > > [W]e conducted additional experiments for distortion with  and found that the quality is indeed comparable or better, which confirms our assumptions:
> >
> > This would be and important observation and I would encourage the author to discuss it in the main text. As $t\leq \log_2 (d)$ are all valid choices, it would make the analysis complete by including all valid $t$ (as the authors uses $d=10$, the set of valid $t$  {$0,1,2,3,4$} is actually very small), whereas only $t=2$ is provided in the response.
> >
> > > Thus, we agree that the term "universal" can be misleading here.
> >
> > It would be beneficial to include related discussion and a more proper claim in the main text.
> >
> > I keep my score intact.

---

> > > ### Author Response · Authors · 2021-09-01
> > > **Additional results for t = 3**
> > >
> > > Indeed, all $t \leq \log_2(d) = \log_2(10) \sim 3.3$ are valid choices, i.e., we can take any $t \in \{0, 1, 2, 3\}$ (note that $t=4$ does not fit here).
> > >
> > > We agree that for completeness it is better to specify all possible values of $t$ and will add this experiment to the paper. Below we show an extended version of the table with all allowed values of $t$.
> > >
> > >
> > > | Signature  | UCSA312 | CS PhDs | Power | Facebook | WLA6 | EuCore |
> > > | --------------- | -------------- | ------------ | ---------- | ------------ | --------- | ----------- |
> > > | $O_{l1}, t=0$ | 0.00324 | 0.0368 | 0.0281 | 0.0458 | 0.0286 | 0.1141 |
> > > | $O_{l1}, t=1$ | 0.00325 | 0.0300 | 0.0231 | 0.0371 | 0.0272 | 0.1117 |
> > > | $O_{l1}, t=2$ | 0.00296 | 0.0335 | 0.0262 | 0.0309 | 0.0273 | 0.1114 |
> > > | $O_{l1}, t=3$ | 0.00257 | 0.0273 | 0.0209 | 0.0313 | 0.0246 | 0.1098 |
> > > | $O_{l2}, t=1$ | 0.00530 | 0.0328 | 0.0246 | 0.0324 | 0.0278 | 0.1127 |
> > > | $O_{l2}, t=2$ | 0.00596 | 0.0303 | 0.0256 | 0.0312 | 0.0278 | 0.1117 |
> > > | $O_{l2}, t=3$ | 0.00303 | 0.0343 | 0.0240 | 0.0302 | 0.0279 | 0.1119 |

---

> > > > ### Comment · Reviewer_ndZi · 2021-09-04
> > > > **Response to Authors**
> > > >
> > > > I would thank the authors for providing the suggested details. I believe the related discussion would be enriching the empirical analysis of the proposed method.

---

### Official Review · Reviewer_VsLk · 2021-07-16

**Rating:** 7
**Confidence:** 2

**Summary:**

Paper proposes overlapping spaces for embedding structured data. Those spaces combine Hyperbolic, Spherical and Euclidean spaces, that all can catch some different specificities of the data. In some sense, it extends the concept of product spaces, allowing some coordinates to be shared between the different spaces

**Limitations And Societal Impact:**

There is no discussion about the limitations and potential negative societal impact.

**Main Review:**

The paper defines overlapping spaces, proposes an optimization scheme and gives an complete experimental evaluation.
The work is original, provides non trivial results and is supported by experimental evidence.

My main concerns are about the writing of the paper that may be improved and about some claims that should be provided with stronger evidence.
The paper is sometimes difficult to read; the paragraph dedicated on the description of paper [9] should be developed more clearly and more details should be provided as the proposed method strongly builds on it.
The description of the overlapping spaces is also quite hard to follow; Fig. 1 helps the reader to figure out the main steps of the method. The links between the curvature and the weights should be given with more details.
Again, in the optimization section, the procedure that is used instead than RSGD should be supported by stronger evidence.
The results provided in the experimental sections support the claims that the new space is definitely useful in a wide set of configurations ; the definition of the WLA6 is interesting. Nevertheless, it would be useful to recall all the notations that are used on the different tables.

That being said, I think that it is a good paper that extends in an appropriate way the product space, allowing to obtain better results.


Minor comments:
- l56: "mertic" should read metric
- the vectors are split in different halves (according to t): what happens if one shuffle the position of the dimensions? Would it be possible to take no adjacent dimensions?

**Time Spent Reviewing:**

4

---

> ### Author Response · Authors · 2021-08-10
> **Authors' response**
>
> Thank you very much for the positive feedback and suggestions. First of all, we will take the recommendations into account and add more details when describing product spaces and overlapping spaces. Let us now address other specific comments.
>
> - **Again, in the optimization section, the procedure that is used instead than RSGD should be supported by stronger evidence.**
>
>     In this regard, we compared the results of our approach with the original implementation provided by Gu et al. In their implementation, RSGD was used. We obtain similar results for the case of distortion and much better results for mAP (better mAP is caused by the fact that we optimize a more suitable objective, so only the distortion results are comparable). A detailed comparison can be found in Supplementary B.2. From this, we concluded that our replacement of RSGD is permissible. Moreover, as we wrote in L198-L201, RSGD is not acceptable for experiments similar to the experiments in Section 5.3. Additionally, according to our comparisons of the two implementations, our approach converges faster in real time.
>
> - **What happens if one shuffle the position of the dimensions? Would it be possible to take no adjacent dimensions?**
>
>     Unfortunately, we are not sure if we understood the question correctly. If not - we will be happy to refine our answer.
>
>     In our universal signature, we choose particular subsets of coordinates. Yes, it is theoretically possible to select any subsets, including ones with non-adjacent coordinates.

---

> > ### Comment · Reviewer_VsLk · 2021-08-30
> > **Thanks for your response**
> >
> > Thanks for your response.
> >
> > I keep my score as I think that the paper is a good paper, with a sound and effective method.
> >
> > I would suggest to rewrite some claims to give more details about why it works and when one should use the space (as stated by rev. oDDg). I would also suggest to rewrite some parts of the paper with additional details about the product and overlapping spaces to make it easier to read.
> >
> > Best,

---

> > > ### Author Response · Authors · 2021-09-01
> > > **Thank you**
> > >
> > > Thank you for your review and valuable suggestions for improving the paper.

---

### Official Review · Reviewer_oDDg · 2021-07-16

**Rating:** 5
**Confidence:** 4

**Summary:**

This paper focuses on graph embedding task and propose the overlapping space that could combine several kinds of spaces in an efficient manner. The proposed method allows subsets of coordinates to be shared between different kinds of spaces. The empirical studies clearly show its promising performance.

**Limitations And Societal Impact:**

The authors have adequately discussed these points.

**Main Review:**

Strengths:
+ The efficiency of combining different spaces is a crucial problem.
+ Experimental results of this paper are promising.

Concerns:
- The motivation of this paper should be discussed more clearly. The authors should explain why we use such complex spaces for learning graph embedding and what kind of data these methods are suitable for.
- The organization of this paper should be improved. The background and related work could be introduced in a more concise way.
- The meaning of Fig. 1 seems not clearly demonstrated. There are too much details in this figure, which hinders the expression of its core information.


**Time Spent Reviewing:**

4

---

> ### Author Response · Authors · 2021-08-10
> **Authors' response**
>
> Thank you for the feedback. We will take the comments into account while preparing the revised version of the paper. If there are more questions or concerns, we will be happy to address them.
>
> - **The motivation of this paper should be discussed more clearly.**
>
>     We briefly describe the motivation in Sections 1 (lines 45-47) and 5.3. In the tasks of information search and recommendations, there are millions and billions of objects, among which one needs to quickly search for the most relevant objects using embeddings built with Siamese models such as DSSM. Because of this, there is a request for sufficiently short vectors that allow storing them all in the memory of a single server. Further, as discussed in Sections 5.4 and A.2, most of these problems have an explicit (recommendations) or implicit (category tree) graph structure, which allows us to talk about the applicability of graphs to these problems. We will write about this more explicitly in the paper.
>
> - **The organization of this paper should be improved. The background and related work could be introduced in a more concise way.**
>
>     Thank you, we will work on that. If you have any particular suggestions, we will be happy to address that.
>
> - **The meaning of Fig. 1 seems not clearly demonstrated. There are too much details in this figure, which hinders the expression of its core information.**
>
>     We will add more intuitive illustrations to the supplementary materials.

---

> ### Author Response · Authors · 2021-08-26
> **We look forward to further discussions**
>
> Thanks again for your review! We improved the paper accordingly. Since there is about a week left to continue the discussion, we would like to know whether you have any additional questions or concerns after reading our response. We look forward to further discussions to address your comments and concerns.

---

> > ### Comment · Reviewer_oDDg · 2021-08-29
> > **Thank you for your response but I still have some concerns**
> >
> > Thank you for your response. The clarification of the motivation is addressed, while I still have some other concerns.
> >
> > First, as is said in the conclusion, the conventional dot product outperforms the complex product space, so it is hard to say the proposed method is empirically effective.
> >
> > Second, as Reviewer VsLk commented, ‘the writing of the paper that may be improved and about some claims that should be provided with stronger evidence’. This is also my major concern. The authors also provide no theoretical discussion to explain why this space is better than others. In my opinion, the author should explain when and to what extent this method works.
> >
> > Meanwhile, if the insights could be expressed with formulas, the whole paper will be much clearer. For example, the optimization process in Sec.4 is totally described by texts without any formula or pseudo-code, which makes the procedure hard to follow.

---

> > > ### Author Response · Authors · 2021-09-01
> > > **Authors' response**
> > >
> > >
> > > Thank you for your response and additional comments. We now address your concerns as follows.
> > >
> > > > First, as is said in the conclusion, the conventional dot product outperforms the complex product space, so it is hard to say the proposed method is empirically effective.
> > >
> > > Please note that in the conclusion we write that "the conventional dot product often outperforms the best product space". However, the dot product is outperformed by the proposed OS-Mixed approach in most experiments.
> > >
> > > (We pay special attention to the comparison with the dot product since it is a strong baseline that is often overlooked in research papers introducing complex metric approaches.)
> > >
> > > > some claims that should be provided with stronger evidence
> > >
> > > As we replied to other reviewers, their comments regarding notation, the choice of the name "universal signature", and an obvious but missing proof of Statement 1 will be corrected.
> > >
> > > > no theoretical discussion to explain why this space is better than others
> > >
> > > As we write in L147, Overlapping Spaces generalize Product Spaces and thus they can give a result no worse than that of a special case. This can be considered as a theoretical justification for the usefulness of our spaces.
> > >
> > > As shown in the experiments, our approach, firstly, gives the best result in many cases, and secondly, it solves the disadvantage of Product Space - brute-forcing the signature, which is an experimental justification for the advantages of our space.
> > >
> > > > Meanwhile, if the insights could be expressed with formulas...
> > >
> > > Comments on the design are accepted and will be corrected, in particular by adding detailed descriptions in the supplement.

---

> > > > ### Comment · Reviewer_oDDg · 2021-09-03
> > > > **Re:**
> > > >
> > > > Thank you for your effort to address my concerns. Overall, I think the idea is good, but this paper needs much improvement, especially writing, before acceptance, so I will keep my score of 5.

---

### Official Review · Reviewer_P7eR · 2021-07-16

**Rating:** 7
**Confidence:** 4

**Summary:**

This paper introduces a model which uses a learned weighted combination of spaces

**Limitations And Societal Impact:**

I feel the authors adequately addressed all limitations of this work, which is of a theoretical nature without direct societal consequences.

**Main Review:**

### Overview

**Originality:** This work is (almost literally) a novel combination of well-known methods, and demonstrates sufficient originality.

**Quality:** The submission is technically sound, however:

- The one "statement" which appears (L145-L146) does not seem to have a corroborating proof in the appendix. (For completeness I would recommend this be included, even if it is relatively straightforward.)
- The authors propose a "universal signature", a name which seems to imply some level of sufficiency / representational capacity, however an explicit proof along these lines is not provided.

That said, the authors perform sufficient empirical analysis of their model, and are very honest about the strengths and weaknesses of their approach.

**Clarity:** The submission is clearly written and well organized, the model should be easily reproducible from their description.

**Significance:** The authors demonstrate improvement on various tasks as well as various aspects of training (see below) of interest to researchers.

---

### Specific Comments / Questions

**General:** The largest objection I could see to this work is that, despite sharing parameterization, these representations are not really "overlapping" in space from a geometric perspective. A critical reviewer could suggest that the performance improvement is simply the result of the model having access to a much richer set of transformations (a contention which is bolstered by the fact that dot product similarity is included, despite not being a metric) and demonstrate this fact by choosing arbitrary collections of otherwise nonsensical functions to include in the set of possible functions to calculate. If the resulting model was still able to perform well, this would point more to the amazing power of optimizers than any sort of combination of spaces with different curvature. Despite this, I think the work itself is interesting and provides many positive contributions to the field.

**L183-L190:** The fact that weights correspond both to changing curvature and different amounts of relative emphasis seems problematic. Does this mean, for example, that the model would struggle with data that, for example, ideally was embedded in mix of a low-curvature Hyperbolic space and high-curvature spherical space?

**L202-L209:** These observations and the proposed solution seem similar to [0], which may be worth citing. As an aside, [0] uses the squared Lorentzian distance, which is similar to the weighted inner product distance except in Hyperbolic space, and it may be of interest to use this in place of (or in addition to) distance on the hyperboloid.

**L239:** (minor) "better than for the best" → "better than the best"

**L250-L251:** (comment) The observation that distortion optimization weakly correlates with mAP optimization is an important one, I appreciate the authors bringing this to the community's attention.

**Section 5.3:** This experiment suggest that neural networks do not seem to easily adapt to overlapping spaces. This could either be a fundamental limitation or simply an area where additional machinery is needed. It seems likely to me, for example, that weight decay may not cooperate nicely with the overlapping spaces, and different forms of regularization may be necessary. In any event, I commend the authors for including this result in the paper itself.

**Section 5.4, 6:** These sections seem a bit rougher, grammatically, than the others.

**Questions / Extensions:**

- I was surprised that an analysis of the weights (eg. how they correlate to the performance of the product space networks and heuristics, how the weights of layer 0 differ when the model is allowed to go to layer 1) were not provided.
- I am also very curious about how the resulting embeddings compare to their single-space and/or product-space equivalents. For example, if we only considered the dot product distance of an OS-Mixed model is it similar to the scores from a trained c-d_dot model? What if we initialize an OS-Mixed model with embeddings from a trained c-d_dot model, and put most of the weight on the dot product portion? (In particular, this seems as though it may allow for better results from the DSSM experiment.) Another avenue for future exploration would be to consider an incremental approach, where (for example) model with t=0 is trained and then expanded to a t=1 model. (To be clear, these are not criticisms of the current paper, which I feel is sufficient, but suggestions for future work.)

[0] Law, Marc, Renjie Liao, Jake Snell, and Richard Zemel. "Lorentzian distance learning for hyperbolic representations." In International Conference on Machine Learning, pp. 3672-3681. PMLR, 2019.


**Time Spent Reviewing:**

4

---

> ### Author Response · Authors · 2021-08-10
> **Authors' response**
>
> Thank you very much for your thorough review and useful suggestions!
>
> - **The one "statement" which appears (L145-L146) does not seem to have a corroborating proof in the appendix.**
>
>     Thank you, we'll add the proof of this statement. It directly follows from the fact that the base distances satisfy the metric axioms and that the aggregations in (6) preserve them.
>
> - **The authors propose a "universal signature", a name which seems to imply some level of sufficiency / representational capacity, however an explicit proof along these lines is not provided.**
>
>     We agree that the term "universal" can be misleading here. In this section, when mentioning universality, we mean that this approach works equally well on all the considered datasets and allows one to effectively use the overlapping spaces in practice without having to select a particular signature for each individual dataset as it happens in product space setup. It is also easy to see that the proposed signature covers a variety of product spaces: each subset of coordinates can be assigned to a space of any fixed curvature.
>
> - **...Does this mean, for example, that the model would struggle with data that, for example, ideally was embedded in mix of a low-curvature Hyperbolic space and high-curvature spherical space?**
>
>     No, since product spaces are an explicit subset of OS, we can learn, e.g., $H_{k} \times S_{d - k}$ signature with the required properties in the same way as it happens in product spaces. Further, if we allow a part of the coordinates to be shared, the remaining ones would still give us flexibility.
>
> - **L202-L209: These observations and the proposed solution seem similar to [0], which may be worth citing**
>
>     Thank you for the reference; we'll add it to the paper. The approach described in [0] differs mainly in that we use Adam instead of Momentum.
>
> - **I was surprised that an analysis of the weights (...) were not provided.**
>
>     While looking at the weights we've made several observations:
>     - We see that OS does not learn a product space. In particular, on the CS PhDs dataset we get ${d_{O_{l=1}, t=0} \propto 0.37 d_H + 0.63 d_S}$, which is significantly better than both $d_S$ and $d_H$ separately.
>
>     - If for $t=0$ there is a space with a noticeably larger weight, then the space of same type often makes the largest contribution for $t = 1$ too. For example, in USCA312, $d_{O_{l1}, t=0} \propto \textbf{0.90} d_E + 0.05 d_H + 0.05 d_S,$ and the weights of the Euclidean subdistances for $d_{O_{l1}}, t=1$ (normalized, $\sum w_i = 1$) are 0.6, 0.15, 0.1.
>
>     -  However, a space that is absent for $t = 0$ can appear for $r = 1$. For example, in the Power dataset, ${d_{O_{l1}, t=0} \propto 0.37 d_H + 0.63 d_S}, {d_{O_{l1}, t=1}(l, r) \propto \textbf{0.1} d_E(l_1^0, r_1^0) + 0.5 d_R(l_1^0, r_1^0) + 0.4 d_H(l_1^1, r_1^1)},$
>         where $l_1^0 = l[0..5], l_1^1 = l[6..10]$.
>
>     - Finally, we noticed that almost always, more than half of the weights are near-zero, which allows one to remove unnecessary distances and improve efficiency.
>
> Regarding other suggestions for improvements, we’ll take them all into account while preparing the revised version of the paper.
>
> Also, thank you for the excellent suggestions for further research, the ideas look very reasonable, and they are partially confirmed by our additional experiments not included in this research. In particular, we noticed that pre-training on a simple space indeed may be used to improve the quality in the DSSM scenario.

---

> > ### Comment · Reviewer_P7eR · 2021-09-02
> > **Analysis of weights and additional observations**
> >
> > Thank you for the clarifications.
> >
> > Personally, I feel like "universal" is perhaps too strong here. Its prior use (eg. "universal approximation theorem") gave me the impression there was something stronger than essentially something defended by empirical results on a range of datasets.
> >
> > Overall, I feel that my original rating for this paper is accurate. Below I outline some replies to your response which I feel would further enhance the submission:
> >
> > - The analysis of the weights is interesting, I think this is certainly worth including in the final paper. I would expand even more on this, ideally with some quantitative experiments which assess how often the observations you've made hold.
> >
> > - What do the weights look like during training? You mention more than half of them are near zero, do you do anything to discourage this during training? If not, it seems possible (likely?) that the random initialization of the spaces essentially determines which weights will dominate - whichever space's score function which is most in alignment with the training objective at initialization will essentially dominate and squash any opportunity for the other spaces to train. Discouraging this would seem to provide a useful mechanism of regularization. Even without the goal of regularization in mind, it seems as though it would be beneficial to provide ample training opportunity for the spaces at the start - perhaps turn off gradients for the weights / add noise to the gradient for these weights at first, and then gradually reduce this during training?

---

> > > ### Author Response · Authors · 2021-09-03
> > > **Authors' response**
> > >
> > > - **The analysis of the weights is interesting, I think this is certainly worth including in the final paper.** This analysis will be added to the final version and we will make it more objective by adding quantitative measurements. Thank you very much for this suggestion, it will improve the paper.
> > > - **...it seems possible (likely?) that the random initialization of the spaces essentially determines which weights will dominate...** The distance weights themselves are initialized uniformly and cannot affect the choice of the final signature. To check the effect of embedding initialization, we performed several runs with different initialization on different graphs ($O_{l1}, t = 1$ on UCSA312, CS PhDs and EuCore) and came to the conclusion that the final signatures turned out to be the same (up to symmetric transformations in the tree) in general, which is why additional experiments with regularization were not carried out. On the other hand, different regularizations here can really be useful in the case of training as part of a larger network, since the optimization problem may be more difficult there, we will consider the possibility of additional experiments here.

---

> > ### Comment · Reviewer_P7eR · 2021-09-03
> > **Additional Clarification Request**
> >
> > I believe all reviewers have maintained their same scores after consideration of the author's rebuttal, and since the average score is still somewhat borderline I felt it was appropriate to reevaluate my assessment of this work. Below, I highlight some concerns that other reviewers have raised, and attempt to understand the objections myself. I ask the authors to clarify any points of misunderstanding on my part, and ask that the other reviewers take these comments into consideration as well.
> >
> > ### Dot Product Outperforming Overlapping Spaces
> >
> > Some reviewers are concerned about the fact that dot product outperforms this method, but from what I see this is not actually frequently the case. In all of Table 2, for example, overlapping spaces seem to perform well in both the metric and similarity classes, taking first in 4/6 datasets and close second for the other two. For Table 3 it seems overlapping spaces are clearly best in the metric case on all but WLA6, where they also present a strong second. For the similarity case, they tie for perfect mAP on 4/6 and outperform on the other two. They also seem to perform best on the WIPS distortion task (Table 6).
> >
> > The only experiments I found where c-dot were better were the higher-dimensional DSSM task (right of Table 4) and metric space bipartite graph reconstruction (though in the similarity case overlapping spaces have better distortion again).
> >
> > **To the authors:** is my understanding of the relative performance on these experiments correct?
> >
> >
> > ### Writing Quality
> >
> > Several reviewers raised concerns with the quality of the writing, and upon further review I have to agree that it is somewhat rough in places. If accepted, I do suggest that a significant amount of effort goes into polishing the writing and the presentation.
> >
> > In particular, reviewers have mentioned that the optimization method in section 4 is unclear. If I understand it correctly (**authors, please clarify my understanding if it is incorrect**) the authors are simply storing Euclidean vectors as the trainable parameters, and leveraging the fact that the spaces under consideration have a global parameterization function from R^d into the space. In a less formal way: you have, for each space of interest $X$, a parameterization which is a function $f_X: \mathbb R^d \to X$. For each embedding you store a set of parameters $P \subset \mathbb R^d$, then given a vector $p \in P$ you first calculate $f_X(p) \in X$ and then calculate the score using the appropriate distances / loss relative to this space. The full model simply takes a weighted combination of these projected losses.
> >
> > This is quite sensible and is the approach I have always taken when in similar settings. I do agree, however, that the writing in this section (particularly) could be made clearer.
> >
> > ### Full Understanding of the Model (Theoretically, Empirically)
> >
> > Several reviewers questions as to whether this paper adequately presents a full understanding of the model. I agree that this is a potential weakness of the work, as it stands currently. It may be difficult to say much, theoretically, about this model which has not already been said about the constituent parts. More precisely, since the model can learn to make all but one weight 0, technically the model can represent anything a single space could represent. The next most reasonable question to consider is whether it is actually possible to train the model to represent this, a question whose answer is well addressed (in the affirmative) by the large set of evaluations performed. What remains, and is admittedly somewhat lacking, are experiments which try to probe the various aspects of the model and understand better what exactly is going on (eg. the experiments regarding the weights I suggested above, which the authors did respond to however a more complete and thorough analysis would be appropriate for the paper). Ideally, a comprehensive work on this proposed model would include such experiments, and this is certainly the largest area for improvement, in my opinion.
> >
> > ### Technical Weaknesses
> >
> > The reviewers pointed out a few technical weaknesses of this work:
> >
> > 1. It is evaluated in relatively small dimensions, and indeed the authors acknowledge that in higher dimensions the performance benefit is not nearly as significant. (Previous experience also suggests to me that it would be quite likely that the model may actually perform *worse* in higher dimensions.) With this awareness, it would be important to emphasize why anyone would prefer this relatively complicated model in lower dimensions instead of simply using (eg.) vector dot product in higher dimensions.
> > 2. I believe the objection I mentioned in my original review still stands, namely that these representations merely share a common underlying parameterization, they are not "overlapping" in space from a geometric perspective, and thus it is possible (and perhaps quite likely!) that the performance improvement is simply the result of the model having access to a much richer set of transformations. I also included a suggested experiment which might dissuade one from this feeling - if I chose some random (smooth, injective) functions as opposed to these geometrically motivated parameterizations and the model performs worse, this is evidence that the choice of parameterization matters, and that somehow it is benefitting from the fact that the chosen parameterization allows access to spaces with different curvature (for example).

---

> > > ### Author Response · Authors · 2021-09-03
> > > **Authors' response**
> > >
> > > - **... is my understanding of the relative performance on these experiments correct?** Yes, you're right. As we understand, the concern of the reviewer oDDg was that the dot product in many cases outperforms the product space (not the proposed overlapping space).
> > > - **...authors are simply storing Euclidean vectors as the trainable parameters, and leveraging the fact that the spaces under consideration have a global parameterization function from $R^d$ into the space...** Yes, your understanding is correct. The concern about writing is accepted and we will improve the text: take into account specific comments of the reviewers and carefully polish the whole text.
> > > - **More precisely, since the model can learn to make all but one weight 0, technically the model can represent anything a single space could represent...** Yes, as we write on L147 any simple or product space is included in the structure as a subset. Our experiments show that when $t$ grows the quality also increases, so the optimization is able to find good configurations.
> > >
> > >     Regarding the general question on model understanding, the analysis of weights will be added to the final version and we will make it more objective by adding quantitative measurements. Thank you very much for this suggestion, it will improve the paper.
> > >
> > > - **...why anyone would prefer this relatively complicated model in lower dimensions instead of simply using (eg.) vector dot product in higher dimensions.** The most straightforward answer is the usage in information retrieval systems, where it is required to store embeddings of billions of objects (similar to Section 5.3) and using small $d$ is important. In our experiments, we choose $d = 10$ just to be consistent with the previous research. Also, for large $d$ (particular values for "large" can depend on an application) the improvements of hyperbolic/product spaces are known to vanish. Regarding our experiments, do you think that adding, e.g., $d = 20$ to our graph reconstruction task will improve the experimental part?
> > > - **... if I chose some random (smooth, injective) functions as opposed to these geometrically motivated parameterizations...**
> > >     This is a good idea for further investigation. We can try to conduct such an experiment and will be grateful for any suggestions of random functions to consider. However, according to our observations, we expect random functions to be unstable. This can be partially evident from Table 2, where, e.g., $c \cdot e^{-\mathrm{dot}}$ has very unstable performance over the datasets. In contrast, the proposed OSMixed approach does not suffer from this instability.

---

> > > > ### Comment · Reviewer_P7eR · 2021-09-10
> > > > **Thank you, I recommend acceptance**
> > > >
> > > > Thank you for the further clarifications.
> > > >
> > > > I believe my rating, as it stands, is correct, and recommend this work be accepted.

---

> > > > ### Comment · Reviewer_P7eR · 2021-09-10
> > > > **More dimensions always helpful but not strictly necessary**
> > > >
> > > > Apologies, I forgot to answer your question when replying above.
> > > >
> > > > I think trying in higher dimensions (eg. $d=20$) would, of course, be interesting, as it's always good to add additional experiments, but I don't think a single additional dimension would add all *that* much to the paper, because it will either outperform or underperform vectors at that point, and we will either learn that the transition point where this occurs is between 10 and 20 or > 10 I am curious about where the performance drop-off starts to occur, however. A plot showing dimension where you actually evaluate every $d=\{10,\ldots, K\}$ where $K$ is large enough that vectors outperform would be of much greater interest, but admittedly is much more work to make rigorous (since both models need to be tuned separately, perhaps, in each dimension).

---

### Decision · Program_Chairs · 2021-09-27

**Decision:**

Accept (Poster)

**Comment:**

This paper had quite a bit of high-quality discussion between reviewers and authors, and is approaching consensus. Multiple reviewers specifically champion the paper and ask for acceptance. The main argument against acceptance, from one of the reviewers is the lack of theoretical analysis. While this is something that would have been nice to have, most of the time, in embedding works, there is no tractable analysis that can be had, without specifying some much more limited version of the problem (e.g., studying distortion bounds for hyperbolic embeddings, but only for trees). So while going off in a theory direction would be interesting, the paper succeeds at its main task.

The paper’s core idea is clever; I can imagine a bunch of work that could take off from seeing this idea. Overall I lean towards acceptance. One thing the authors should work on is some additional clarity in the writing for the final version; this was noted by several reviewers, and I agree.